

# The concurrent validity and reliability of the Leg Motion system for measuring ankle dorsiflexion range of motion in older adults

Carlos Romero Morales[1,*], César Calvo Lobo[2,3,*], David Rodríguez Sanz[1,*], Irene Sanz Corbalán[4,*], Beatriz Ruiz Ruiz[1,*] and Daniel López López[5,*]

[1] Physical Therapy & Health Science Research Group, Physiotherapy Department, Faculty of Health, Exercise and Sport, Universidad Europea de Madrid, Villaviciosa de Odón, Madrid, España

[2] Departamento de Fisioterapia, Centro Superior de Estudios Universitarios La Salle, Universidad Autónoma de Madrid, Spain

[3] Motion in Brains Research Group, Instituto de Neurociencias y Ciencias del Movimiento, Centro Superior de Estudios Universitarios La, Universidad Autónoma de Madrid, Madrid España

[4] School of Nursing, Physiotherapy and Podiatry, Universidad Complutense de Madrid, Madrid, España

[5] Research, Health and Podiatry Unit, Department of Health Sciences. Faculty of Nursing and Podiatry, Universidade da Coruña, Ferrol, A Coruña, Spain

[*] These authors contributed equally to this work.

Corresponding authors
Carlos Romero Morales,
carlos.romero@universidadeuropea.es
César Calvo Lobo,
cecalvo19@hotmail.com
David Rodríguez Sanz,
davidrodriguezsanz@gmail.com
Irene Sanz Corbalán,
iresanzcorbalan@gmail.com
Beatriz Ruiz Ruiz,
beatriz.ruiz@universidadeuropea.es
Daniel López López,
daniellopez@udc.es

## ABSTRACT

**Background**. New reliable devices for range of motion (ROM) measures in older adults are necessary to improve knowledge about the functional capability in this population. Dorsiflexion ROM limitation is associated with ankle injuries, foot pain, lower limb disorders, loss of balance, gait control disorders and fall risk in older adults. The aim of the present study was to assess the validity and reliability of the Leg Motion device for measuring ankle dorsiflexion ROM in older adults.

**Methods**. Adescriptive repeated-measures study was designed to test the reliability of Leg Motion in thirty-three healthy elderly patients older than 65 years. The subjects had to meet the following inclusion and exclusion criteria in their medical records: older than 65 years; no lower extremity injury for at least one year prior to evaluation (meniscopathy, or fractures) and any chronic injuries (e.g., osteoarthritis); no previous hip, knee or ankle surgery; no neuropathic alterations and no cognitive conditions (e.g., Alzheimer's disease or dementia). Participants were recruited through the person responsible for the physiotherapist area from a nursing center. The subjects were evaluated in two different sessions at the same time of day, and there was a break of two weeks between sessions. To test the validity of the Leg Motion system, the participants were measured in a weight-bearing lunge position using a classic goniometer with 1° increments, a smartphone with an inclinometer standard app (iPhone 5S®) with 1° increments and a measuring tape that could measure 0.1 cm. All testing was performed while the patients were barefoot. The researcher had ten years of experience as a physiotherapist using goniometer, tape measure and inclinometer devices.

**Results**. Mean values and standard deviations were as follows: Leg Motion (right 5.15 ± 3.08; left 5.19 ± 2.98), tape measure (right 5.12 ± 3.08; left 5.12 ± 2.80), goniometer (right 45.87° ± 4.98; left 44.50° ± 5.54) and inclinometer app (right 46.53° ± 4.79; left 45.27° ± 5.19). The paired $t$-test showed no significant differences between the limbs or between the test and re-test values. The test re-test reliability results for Leg

Motion were as follows: the standard error of the measurement ranged from 0.29 to 0.43 cm, the minimal detectable difference ranged from 0.79 to 1.19 cm, and the intraclass correlation coefficients (ICC) values ranged from 0.97 to 0.98.

**Conclusions**. The results of the present study indicated that the Leg Motion device is a valid, reliable, accessible and portable tool as an alternative to the classic weight-bearing lunge test for measuring ankle dorsiflexion ROM in older adults.

## INTRODUCTION

Range of motion (ROM) measurements are considered to be an important factor in the physiotherapeutic assessment of the general population and of special groups, such as sportsmen (*Skarabot et al., 2015*) and older adults (*Sacco et al., 2015*). Changes in ankle dorsiflexion ROM have been associated with foot pain, ankle injuries (*Youdas et al., 2009*), neuritis and lower limb disorders (*Young et al., 2013*).

Spain tops the list of the European countries that require a higher rate of geriatric care, and is a clear example of growing elderly population (*Kolb et al., 2011*).

It is known that the aging process in elderly people reduces quality of life and causes modifications in the locomotor system (*Hongo et al., 2007*), reducing the ROM and limiting mobility. A restricted ROM may predispose individuals to a lack of coordination (*Yingyongyudha et al., 2015*), can increase the fall risk (*Gajdosik et al., 2005*), is an excellent predictor of the loss of ambulation in elderly people (*Bakker et al., 2002*) and it is presented in many adult pathologies that affect postural control and gait. Daily activities such as walking, descending stairs, and kneeling require 10° of ankle dorsiflexion ROM (*Harris, 1991*), while other actions such as running require 20° to 30° (*Pink et al., 1994*).

It is very common in the literature to find studies, whose objective is to validate measurement instruments for the elderly population (*Rogan, De Bie & Douwe de Bruin, 2016*; *Yorozu, Moriguchi & Takahashi, 2015*).

In the study of Slavko et al. (*Rogan, De Bie & Douwe de Bruin, 2016*), it is stated that there are several methods to assess gait in humans, but its feasibility has not been demonstrated in frail populations. In the case of ROM measurements, there are several studies that attempt to validate the applicability of different instruments for elderly adults, as reported Sacco and colleagues which aims to validate a measuring instrument in the elbow joint range in gerontology (*Sacco et al., 2015*), although no studies have been found for ankle dorsiflexion.

Based on a reliability dorsiflexion ROM study, the weight-bearing measures are more reliable (ICC = 0.93–0.96) than non-weight-bearing postures (ICC = 0.32–0.72) during practical activities, such as walking, running or stair ambulation (*Venturini et al. 2006b*). Multiple tools, such the inclinometer, goniometer and tape measure, have been used to measure the ankle ROM (*Konor et al., 2012*).

Tape measure is an easy way to determine ankle flexion ROM in the weight bearing position (*Venturini et al., 2006b*). According to *Bennell et al. (1998)* the test starts in the standing position, with the subject's foot on the tape line, perpendicular to the wall. The subject then moves the foot away from the wall until the knee touches the wall lightly without lifting the heel from the ground. The distance between the big toe and the wall (in centimeters) is then measured. A tape measure is a cheap tool that can easily be transported and quickly and safely used in many settings. However, there are some potential variations that occur during testing that need to be controlled. For instance, variations in the subtalar and foot position (*Bohannon, Tiberio & Waters, 1991*) and the visual reference for the knee or the maintenance of the foot alignment during the performance of the test may change dorsiflexion results (*Kim et al., 2011*) are the main limitations with regard to the standardization of this test.

The goniometer is frequently used to evaluate ankle joint dorsiflexion (*Kim et al., 2011*) It is an inexpensive and portable tool, but experience is required for its accuracy and effective use.

Another approach to quantifying the ankle dorsiflexion ROM is an inclinometer; *Venturini et al. (2006a)* reported reliable results when performing measurements with a digital inclinometer (ICC = 0.84–0.95) compared with a goniometer (ICC = 0.65–0.89). Smartphones are accessible and easy to handle devices, with inclinometer apps (*Vohralik et al., 2015*). *Wellmon et al. (2015)* performed a validity and reliability examination of three different smartphones inclinometer apps (iPhone 5®, Samsung SII and LG). The interrater reliability (ICC = 0.995–1.000) and validity (ICC = 0.998–0.999) were excellent. It is only necessary for the implementation of this measurement, finding the tibial tuberosity for stabilizing the smartphone in a weight-bearing position. The inclinometer uses a digital display to report the angle of decline (*Cosby & Hertel, 2011*). Despite the high intraclass correlation coefficient values for the goniometer and inclinometer, there is no universal agreement on the choice of one method over the other (*Konor et al., 2012*).

Check your MOtion (Spain) developed the Leg Motion system, a new, accessible, lightweight and portable tool for evaluating the ankle dorsiflexion ROM in the weight-bearing position (*Calatayud et al., 2015*). During assessment with the Leg Motion system, the big toe is placed at the starting line and the knee touches a metal stick while keeping the foot in the same position without removing the heel from the surface. The metal stick is progressed along the line to the maximal ankle dorsiflexion. Leg Motion shows greater standardization because can be applied on any surface and doesn't need the presence of a wall during the measurement. Therefore, Leg Motion is considered to be a portable tool that can be tested in any location or on any surface, "where a measuring tape needs to be placed or where the normal weight-bearing lunge test has limitations, for example: the variations in the subtalar and foot position during the measurement," according to *Calatayud et al. (2015)*, who provided evidence to support the use of the Leg Motion device in healthy subjects.

The aim of the present study was to test the validity and reliability of Leg Motion for measuring ankle dorsiflexion ROM in older adults.

## METHODS

### Study design

A descriptive repeated-measures study was performed between April and June 2015.

### Participants

Thirty-three healthy elderly subjects (age 71 ± 3.6 years, height 167.0 ± 10 cm, weight 68.24 ± 13.47 kg, and body mass index 24.31 ± 3.50 kg/m$^2$) were included in the study. Before starting the procedure, all the participants read and signed an institutional informed consent. All protocols used in the study comply with the items listed in the 1975 Declaration of Helsinki and its 2008 review.

Participants were recruited through the person who was responsible for the physiotherapist area at a care center. The subjects had to meet the following inclusion and exclusion criteria: over 65 years of age; no lower extremity injury for at least one year prior to the evaluation (i.e., meniscopathy, fractures), and any chronic injuries (i.e., osteoarthritis); no previous hip, knee or ankle surgery; no neuropathic alterations and no cognitive conditions (i.e., Alzheimer's disease and dementia). The Scientific Committee of the European University of Madrid (CIPI/048/15) approved this study.

### Procedure

In accord with the protocol established by *Calatayud et al. (2015)* the subjects were evaluated in two different sessions at the same time of day and there was a break of two weeks between sessions.

The participants were measured in a weight-bearing lunge position using a classic goniometer with 1° increments (Baseline, Yarmouth, ME, USA), a smartphone with an inclinometer standard app (iPhone 5S®) with 1° increments and a measuring tape that could measure 0.1 cm. All testing was performed while the patients were barefoot and three trials were performed for each leg, in each testing method. The researcher had ten years of experience as a physiotherapist and using a goniometer, tape measure and inclinometer.

Leg Motion is a new device; therefore, the physiotherapist had only one and a half years' experience with this tool.

The tape measure protocol was performed with the participants in the standing position with their heels touching the ground, knees aligned with the second toe, and the big toe 10 cm away from the wall (Fig. 1).

The participants could be lightly supported on the wall using the index and middle fingers of each hand. As proposed by *Konor et al. (2012)*, the subjects were encouraged to move their knees toward the wall (maintaining alignment with the second toe) until their knees touched the wall. The foot continued to move away from the wall 1 cm at a time, and the participants performed this exercise again until they were not able to contact the wall with their knees without lifting their heels off the surface. From that point on, the foot moved to the nearest 0.1 cm increments away from the baseline until the knee contacted the wall (*Hoch & McKeon, 2011*).

Maximal dorsiflexion ROM during measuring tape test was defined as the maximum distance of the toe from the wall, while maintaining contact between the wall and knee,

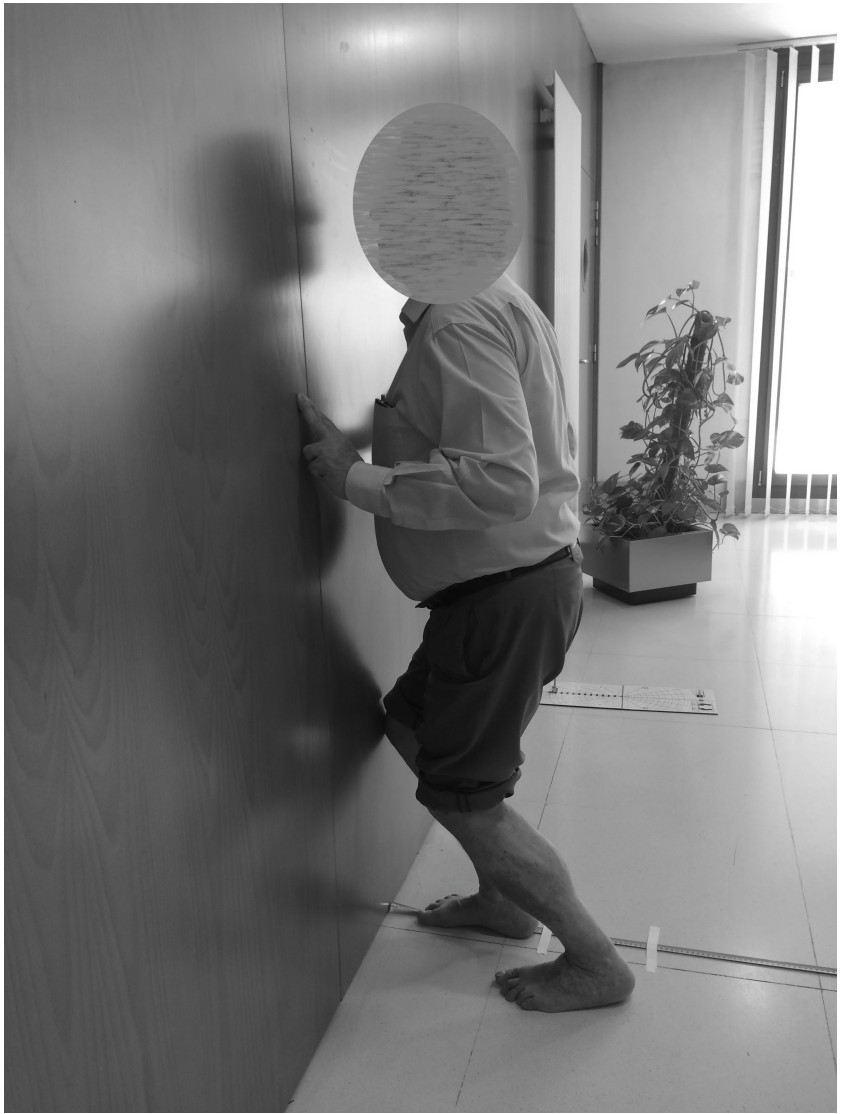

**Figure 1** **Position for Tape Measure protocol.**

without lifting the heel (*Konor et al., 2012*). This thorough procedure was performed to ensure accuracy to the nearest millimeter (*Hoch & McKeon, 2011*).

Once the patient reached the final lunge position at maximal dorsiflexion, a classic goniometer was aligned with the fibula (mobile branch) and the floor (stable branch) (Fig. 2) (*Konor et al., 2012*).

After the goniometer measurement, in the same position, an iPhone 5S® with an inclinometer app was placed at the tibial tuberosity to evaluate the angle between the tibia and the ground (Fig. 3) (*Konor et al., 2012*).

Finally, according to the procedure by *Calatayud et al. (2015)* for the Leg Motion device, patients were in a standing position with their test foot on the evaluation scale (Fig. 4) (Fig. 5). In this position, the participants performed a lunge in which the knee was flexed
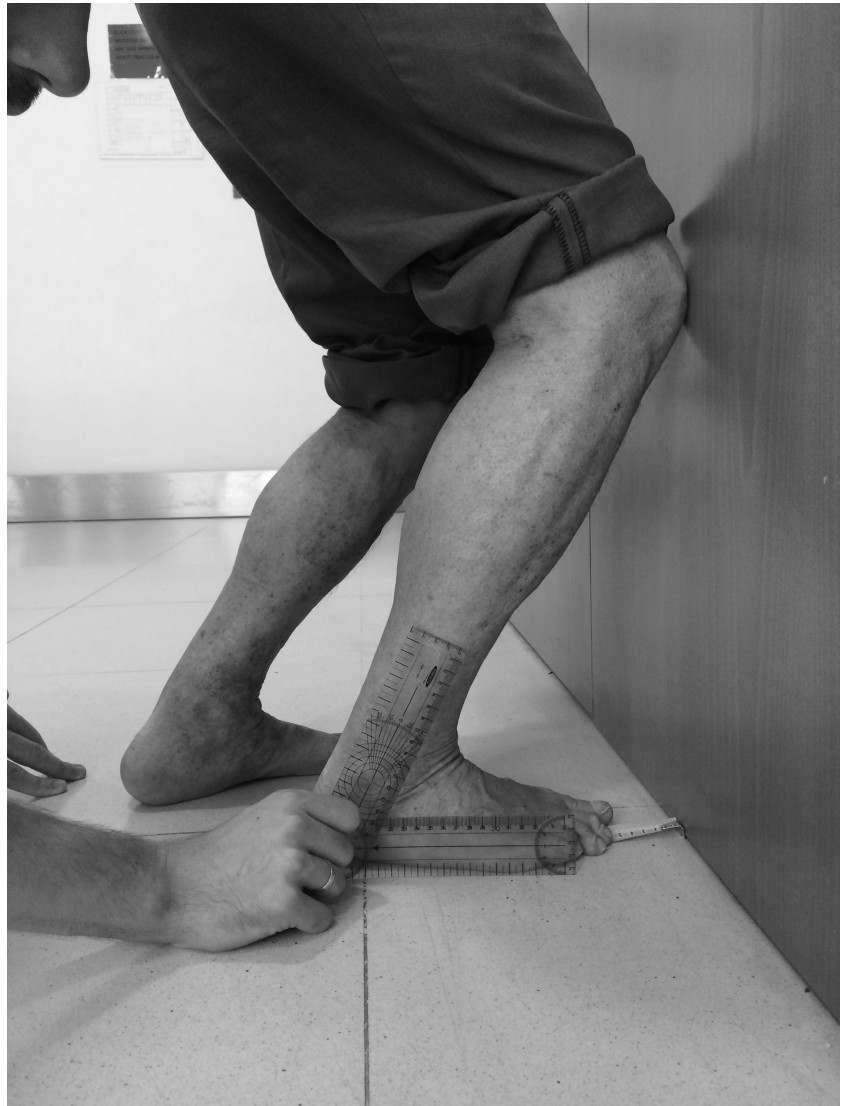

**Figure 2** **Goniometer measurement at the final lunge position in maximal dorsiflexion position.**

in order to facilitate contact between the anterior knee and a metal stick. If the patient was able to maintain contact with the stick for three seconds without lifting their heel off the surface, the metal stick was progressed away from the knee. As specified by developers, the "Leg Motion system test was defined as the maximum distance between the toe to the metal stick where contact between the stick and the knee was maintained without lifting the heel for three seconds" (*Calatayud et al., 2015*).

Three trials were performed for each leg. The first was performed with one leg during three seconds, and then with the other in a counterbalanced order; thus prevents muscle fatigue, by alternating measurements between both legs; the mean value of the three trials was used for data analysis. All the procedures were performed while the subject was barefoot. If the participants did not meet any of the standards described for the test, they had to repeat the trial.

Romero Morales et al. (2017), *PeerJ*, DOI 10.7717/peerj.2820     **6/14**

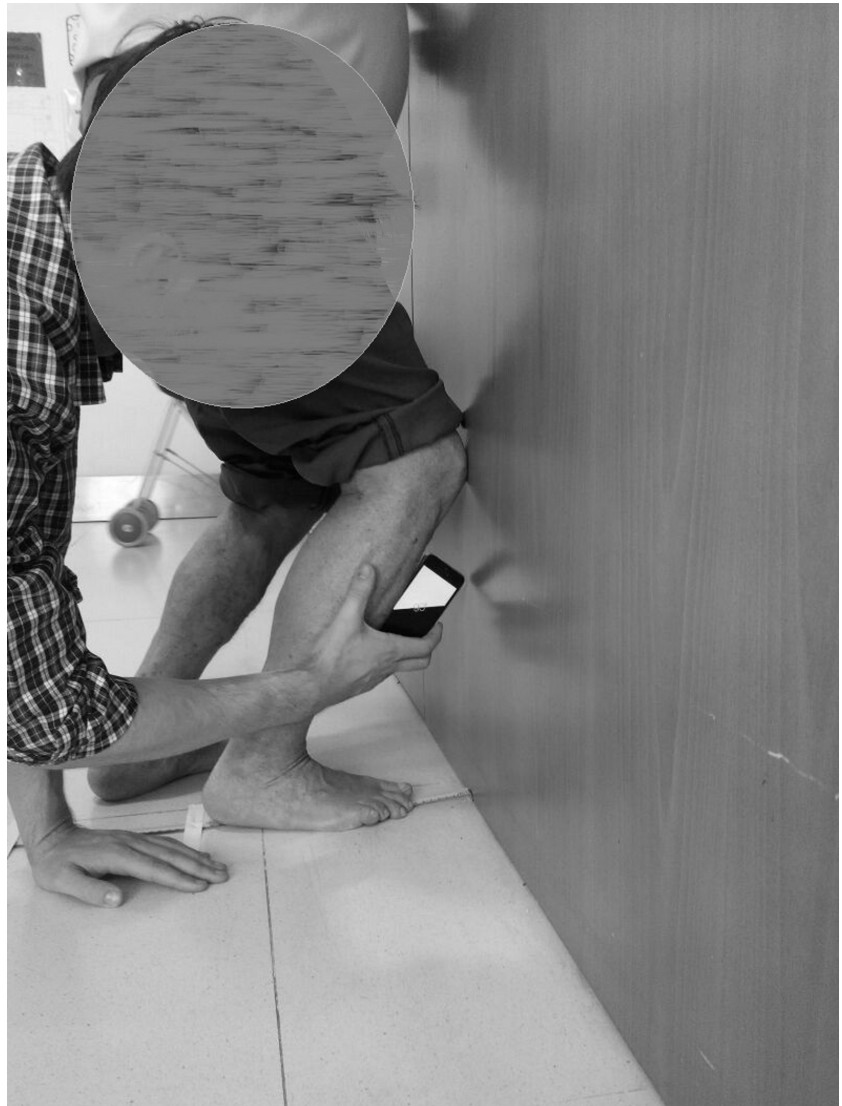

**Figure 3** Procedure to determine the tibial tuberosity to evaluate the angle of the tibia to the ground with inclinometer app.

## Statistical procedure

SPSS version 22.0 for Windows was used for statistical analysis. Descriptive statistics were determined for each measurement. The mean and standard deviation (SD) were calculated for both limbs. A paired $t$-test was performed to establish significant differences in the scores obtained at test and retest sessions.

The intra-rater reliability was determined using ICC. The standard error of measurement (SEM) and the 95% confidence intervals (CI) were calculated to estimate the error associated with the measurement *Weir (2005)*. The reliability was defined as poor (ICC < 0.50), moderate (ICC 0.50 to 0.75), and good (ICC > 0.75) (*Portney & Watkins, 2009*). In addition, the minimum detectable change (MDC) was calculated, based on the standard

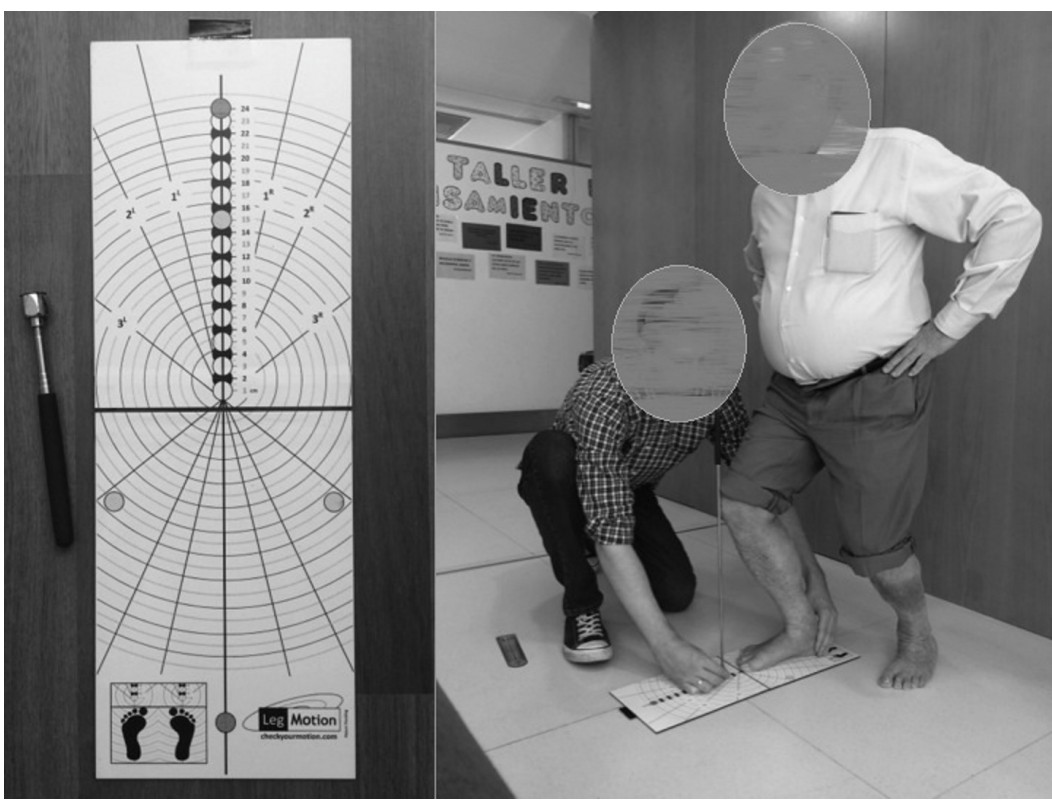

**Figure 4** **Leg Motion system and procedure.**

error of measurement (SEM), by the formula SEM*1.96*$\sqrt{2}$, in order to avoid the error range of instrument measurement (*Weir, 2005*).

To assess the relationship between Leg Motion and other ankle dorsiflexion ROM measures, we performed a Pearson correlation analysis.

## RESULTS

The mean values and SD were as follows: Leg Motion of the right leg was 5.15 ± 3.08 and on the left was 5.19 ± 2.98; by tape measure was 5.12 ± 3.08 on the right and 5.12 ± 2.80 on the left; by goniometer was 45.87° ± 4.98 on the right and 44.50° ± 5.54 on the left and by the inclinometer app was 46.53° ± 4.79 on the right and 45.27° ± 5.19 on the left (Table 1).

The correlation coefficients between Leg Motion and other ankle dorsiflexion ROM measurements are presented in Table 2.

The paired $t$-test showed no significant differences between the limbs and the test and re-test values. The test re-test reliability results for Leg Motion were as follows: SEM ranged from 0.29 to 0.43 cm, MDC ranged from 0.79 to 1.19 cm, and ICC values ranged from 0.97 to 0.98 (Table 3).

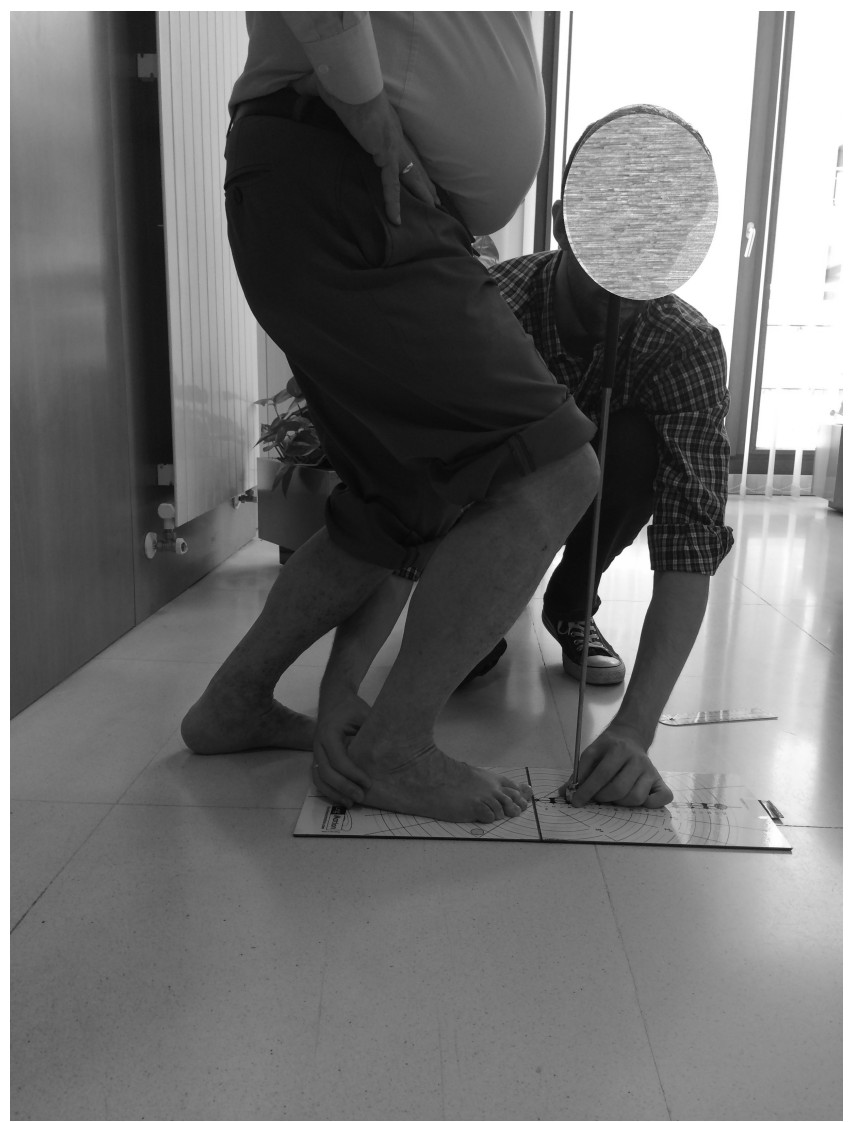

**Figure 5   Leg Motion procedure.**

**Table 1   Results of ankle dorsiflexion range of motion measurements.**

| Test | Side | Mean ± SD |
| --- | --- | --- |
| Leg Motion | Right side | 5.15 cm ± 3.08 |
| | Left side | 5.19 cm ± 2.98 |
| Tape measure | Right side | 5.12 cm ± 3.08 |
| | Left side | 5.12 cm ± 2.80 |
| Goniometer | Right side | 45.87° ± 4.98 |
| | Left Side | 44.50° ± 5.54 |
| Inclinometer app | Right side | 46.53° ± 4.79 |
| | Left side | 45.27° ± 5.19 |

**Table 2** Correlation coefficients between Leg Motion results and other ankle dorsiflexion range of motion measurements.

| Test | Side | Leg Motion right | Leg Motion left |
|------|------|------------------|-----------------|
| Tape measure | Right side | 0.97[*] | 0.86[*] |
| | Left side | 0.87[*] | 0.99[*] |
| Goniometer | Right side | 0.78[*] | 0.78[*] |
| | Left side | 0.51[*] | 0.65[*] |
| Inclinometer app | Right side | 0.02 | <0.01 |
| | Left side | 0.50[*] | 0.49[*] |

**Notes.**
  [*]Significant Pearson's correlation coefficients ($p < 0.01$).

**Table 3** Intrarater reliability for Leg Motion results on measurement.

| Side | Mean | SD | SEM | MDC | ICC (95% CI) |
|------|------|-----|-----|-----|--------------|
| Right side | 5.11 cm | 3.04 cm | 0.43 cm | 1.19 cm | 0.97 (0.94; 0.98) |
| Left Side | 5.19 cm | 2.85 cm | 0.29 cm | 0.79 cm | 0.98 (0.95; 0.99) |

**Notes.**
  Abbreviations: SD, standard deviation; SEM, standard error of measurement; MDC, minimal detectable difference; ICC, intraclass correlation coeficient; CI, confidence intervals.

## DISCUSSION

This study showed the reliability of Leg Motion to measure ankle dorsiflexion ROM in older adults. Highly reliable results were shown for the test-retest measures because SEM values ranged from 0.29 to 0.43 cm and ICC ranged from 0.97 to 0.98. Furthermore, the validity was stablished because MDC ranged from 0.79 to 1.19 cm. Nevertheless, it is valid due to the high correlation with the established methods both the MDC and the absolute value. Therefore, its reliability was confirmed, so the Leg Motion could be an alternative choice to the classic weight-bearing test for the measurement of ankle dorsiflexion ROM in elderly people. The advantage of using the Leg Motion over the tape measure using the distance-to wall technique is that the subject does not have to move the foot, so there aren't variations in the subtalar and foot position, or changes of the foot alignment during the performance of the test which could change dorsiflexion results (*Kim et al., 2011*).

In the literature, only one study was found that used this tool to evaluate university students, and the authors had similar ICC values (right side 0.98; left side 0.96) for the intra-rater reliability using the Leg Motion system (*Calatayud et al., 2015*). These findings support the authors' hypothesis that Leg Motion is a reliable and valid tool for measuring the ankle dorsiflexion ROM in elderly adults. Moreover, *Calatayud et al. (2015)* found statistically significant Pearson's coefficients ($p < 0.01$) between the Leg Motion test and a tape measure, goniometer and inclinometer. This study had similar results ($p < 0.01$) for the tape measure and goniometer in both limbs in this specific population.

On the other hand, with respect to the inclinometer app results, the authors found significant Pearson's correlation coefficients on the left side (ranged from 0.49 to 0.50), but not on the right side (ranged from 0.00 to 0.02). In contrast to this study, *Vohralik et al. (2015)* examined the iHandy Level app for the ankle dorsiflexion ROM using a

weight-bearing lunge test. The test showed excellent intra-rater reliability (ICC values ranged from 0.76 to 0.97), indicating that this app is reliable and valid for measuring ankle dorsiflexion ROM.

Wellmon et al. (2015) suggest that there are errors inherent to measurement with smartphone apps due to patient factors, app domain and examiner skills. In this study, the muscle fatigue, the loss of balance and an altered biomechanic by age can also be predisposing factors, which may explain the absence of significant Pearson's correlation coefficients on right side for the inclinometer app.

The SEM and MDC results in the current study were greater than the values provided by Calatayud et al. (2015) (intra-rater SEM ranging from 0.58 to 0.80 cm; intra-rater MDC ranging from 1.60 to 2.23 cm) for Leg Motion measurements.

A decreased ankle dorsiflexion ROM increases instability (Cruz-Díaz et al., 2015), which directly impacts the loss of balance and increases the fall risk in older adults (Schiller, Kramarow & Dey, 2007). Therefore, it is necessary to help the elderly maintain their ability to walk, climb stairs and control their gait. The number of falls is increasing worldwide in older adults (Williams et al., 2015). Therefore, it is necessary to establish prevention programs to reduce the fall risk.

Leg Motion could be very useful for monitoring the ankle dorsiflexion ROM values and as a training and prevention tool. Moreover, in the study conducted by Calatayud et al. (2015), the metal stick can be a visual target to maintain foot and knee alignment and facilitate more tactile stimulus for the patient, in order to improve the right dorsiflexion execution. On the other hand, our study subjects had no disturbances of balance, although it is known that old age carries a lower engine and progressive loss of balance control. Maybe we could have controlled this possible fear of falling; placing the Leg Motion in parallel bars, for example. We consider this aspect a limitation of the study that is easily solvable for future researches.

Another limitation of this study is that healthy participants were studied; therefore, the results cannot be extrapolated to other altered populations. Additionally, an important limitation is the appearance of fatigue after standing without rest for a few minutes. Further studies are necessary to improve knowledge about this device in older adults with different conditions.

## CONCLUSIONS

The Leg Motion system gives a visual and tactile stimulus that allows an adequate execution of the dorsiflexion measurement, avoiding variations in the foot position. The results of the present study indicated that the Leg Motion device is a valid and reliable tool that can be used as an alternative to the classic weight-bearing lunge test for measuring ankle dorsiflexion ROM in older adults.

## ACKNOWLEDGEMENTS

We thank the patients who participate in this research.

### Funding
The authors received no funding for this work.

### Competing Interests
The authors declare there are no competing interests.

### Author Contributions
- Carlos Romero Morales, César Calvo Lobo, David Rodríguez Sanz, Irene Sanz Corbalán, Beatriz Ruiz Ruiz and Daniel López López conceived and designed the experiments, performed the experiments, analyzed the data, contributed reagents/materials/analysis tools, wrote the paper, prepared figures and/or tables, reviewed drafts of the paper.

### Human Ethics
The following information was supplied relating to ethical approvals (i.e., approving body and any reference numbers):

The Scientific Committee of the European University of Madrid (Spain) (CIPI/048/15) approved this study.

### Data Availability
The raw data has been supplied as a Data S1.

### Supplemental Information
Supplemental information for this article can be found online at http://dx.doi.org/10.7717/peerj.2820#supplemental-information.

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
