# Peer review of "The concurrent validity and reliability of the Leg Motion system for measuring ankle dorsiflexion range of motion in older adults"

_PeerJ, doi:10.7717/peerj.2820_

## Round 0.1 · original submission · Major Revisions

The two reviewers have highlighted a number of issues with the initial submission of this manuscript that require careful attention. I would therefore suggest you carefully read these comments and consider how you can improve the manuscript to the standard required by the two reviewers.

Reviewer 1 ·

Basic reporting

Please see comments within the uploaded document

Experimental design

Please see comments within the uploaded document

Validity of the findings

Please see comments within the uploaded document

Additional comments

The key components of the manuscript are there, however I don't believe this study adds a great deal of novel and new information. The manuscript did not illustrate how the leg motion system is superior to the weight bearing lunge test. I question the practicality, safety, and cost of this new device. Considering that you would use this on older adults I would recommend a wall etc. to provide support if balance deficits are apparent (it is claimed that the leg motion system is superior as no wall is needed). I believe this is not an advantage over existing methods.
There are also several grammatical flaws within this manuscript (see uploaded document with comments inserted).

A great deal of work is needed within the manuscript (introduction and discussion) to show the need for this new device and how it is superior to the current available methods.
The conclusion is very brief and does not summarize the study well. Please work on this further.

Annotated reviews are not available for download in order to protect the identity of reviewers who chose to remain anonymous.

·

Basic reporting

This is an interesting article and I think it has value if the authors can provide a more thorough justification for the need of this research.

It is unclear from the introduction the purpose for repeating this study in elderly populations. Details have been provided for the use of DF ROM measurements in elderly. The reliability of other measurements has been detailed but the need for another tool and the reasons for needing to check reliability in elderly populations has not be provided. The authors should consider reducing the focus on the reliability of other measurements and focus on factors that make other measurement techniques inconvenient or less reliable and also factors that may influence reliability of the measurement in elderly populations compared to the population previously assessed. The purpose for measuring three other methods is not clear.

Experimental design

Method – curious as to how fatiguing this protocol might have been for the test population. Reproducing multiple squatting motions. Did they then have to hold this position while all three established measurements were taken? Or did they stand up and then squat again for each measurement? Would this have introduced variability into the measurements.

Line 149 – small increments – please specify and what is baseline? I’m assuming this is the last distance where they were successful in touching the wall with knees without taking heels off. Please provide more details in this sentence.

Line 174 – was three trials for each leg performed for all tests or just the Leg Motion testing? Please specify.

Line 176 – Do you need to state this again. You already said they were shoeless in line 136. I would state they were barefoot one at the beginning (rather than shoeless as this does not directly imply barefoot.)

Line 182 – “A paired t-test was performed to avoid significant differences in the scores obtained at test and retest sessions.” Avoid not the correct term here. Establish might be better?

Line 187 – “Additionally, minimal detectable differences (MDD) were analysed (SEM*1.96*√2) and controlled so that the measurements were real and outside the error range”. Suggest rewording. Not clear on what the author is trying to say. How were they controlled? Why were they controlled?

Validity of the findings

Reliability as reported as SEM and ICC are slightly different to MDD and would make more sense reported separately to these measures. In the sentence starting line 212 has combined all three measures as justification for concluding the measure to be reliable. The following sentence then concludes the measure to be valid by referring back to the reliability results. I would suggest that the MDD is more associated with validity, however the comparison with the established methods both the MDD and the absolute value is important in establishing the validity of the measurements.
Suggest rewording the first paragraph to clearly separate the two concepts.

Line 219 – I would suggest including – in elderly adults – into the sentence as this is what is being tested and I am assuming that there is some justification as to why the authors think there may be a difference in reliability in this population.

Paragraph starting line 229 – consider that fatigue from the measurement procedure may also have an impact on these measurements in this particular patient population.

Additional comments

Line 138 – “therefore the physiotherapist was only one year and a half experience with this tool.” Change to - therefore the physiotherapist had only one and a half years’ experience using this tool.

Line 145 – change “As proposed as ….” To As proposed by….

Line 240 – correct aduls to adults

Line 244 – overcome stimulus – not sure what the authors mean here. Please clarify this sentence.

---

## Round 0.2 · Minor Revisions

We thank you for addressing most of the comments from the reviewers. Please make the final suggested minor changes from reviewer 2 and make it a priority to double check for grammatical errors that may still be contained in the revision.

·

Basic reporting

No comment

Experimental design

No comment

Validity of the findings

No comment

Additional comments

I am happy with the additions made to the paper in response to my review. There are still some grammatical errors that need to be addressed but the reason for introducing another tool has been justified.

Sentence starting line 77. Is very long. Suggest ending the sentence at line 80 “affect postural control and gait.”
Suggest rewording the following lines “so that having a correct ankle ROM assessment it is especially important in the case of elder adults, due to this natural loss of gait control in this kind of population, directly in relation with the ankle function.”

Line 220 – correct stablish to establish

---

## Round 0.3 · accepted · Accept

We thank you for your diligent work in replying to the editor's comments. We are now happy to accept the manuscript for publication.